# An institutional analysis of graduate outcomes reveals a contemporary workforce footprint for biomedical master's degrees

Jack Hart[1], Caleb C. McKinney[1,2]*

1 Biomedical Graduate Education, Georgetown University Medical Center, Washington, District of Columbia, United States of America, 2 Department of Rehabilitation Medicine, Georgetown University Medical Center, Washington, District of Columbia, United States of America

* caleb.mckinney@georgetown.edu

**Data Availability Statement:** All relevant data that maintains anonymity are within the manuscript and its Supporting Information files.

## Abstract

There is continued growth in the number of master's degrees awarded in the life sciences to address the evolving needs of the biomedical workforce. Academic medical centers leverage the expertise of their faculty and industry partners to develop one to two year intensive and multidisciplinary master's programs that equip students with advanced scientific skills and practical training experiences. However, there is little data published on the outcomes of these graduates to evaluate the effectiveness of these programs and to inform the return on investment of students. Here, the authors show the first five-year career outlook for master of science graduates from programs housed at an academic medical center. Georgetown University Biomedical Graduate Education researchers analyzed the placement outcomes of 1,204 graduates from 2014–2018, and the two-year outcomes of 412 graduates from 2016 and 2017. From the 15 M.S. programs analyzed, they found that 69% of graduates entered the workforce, while 28% entered an advanced degree program such as a Ph.D., allopathic or osteopathic medicine (M.D. or D.O.), or health professions degree. International students who pursue advanced degrees largely pursued Ph.D. degrees, while domestic students represent the majority of students entering into medical programs. Researchers found that a majority of the alumni that entered the workforce pursue research-based work, with 59% of graduates conducting research-based job functions across industries. Forty-nine percent of employed graduates analyzed from 2016 and 2017 changed employment positions, while 15% entered advanced degree programs. Alumni that changed positions changed companies in the same job function, changed to a position of increasing responsibility in the same or different organization, or changed to a different job function in the same or different company. Overall, standalone master's programs equip graduates with research skills, analytical prowess, and content expertise, strengthening the talent pipeline of the biomedical workforce.

**Funding:** The author(s) received no specific funding for this work.

**Competing interests:** The authors have declared that no competing interests exist.

## Introduction

In a time of increasing competition for funded biomedical STEM Ph.D. slots, and a narrowing bottleneck in the professoriate pipeline for Ph.D.'s [1–4], educational institutions must recalibrate their metric for success in graduate education. In recent years, Ph.D. conferrals have stagnated and Ph.D. recipients utilize their research skills across employment industries within and outside of academia [5, 6]. This trend is concomitant with a steady increase in the number of master's degrees being awarded annually in the life sciences. While career outcome taxonomies and workforce trends have been studied for doctoral graduates and professional master's graduates [7, 8], outcomes for M.S. degree recipients have not been elucidated.

According to the 2018 Graduate Enrollment and Degrees Report by the Council of Graduate Schools, the growth in awarded doctoral degrees for life sciences has decreased to a crawl [9]. More explicitly, the number of doctoral degrees awarded from 2016–17 to 2017–18 increased only by 0.6%, compared with the 2.1% average annual increases seen between 2007–08 and 2017–18. Moreover, this recently slowed growth in life science doctoral degrees is in stark contrast to the 2.4% growth in master's degree production from 2016–17 to 2017–18.

These trends from the CGS appear to be corroborated by the National Science Foundation (NSF). Based on our analysis of the NSF data on awarded advanced degrees in the biological sciences between 2005 and 2015 [10], the number of awarded master's degrees have been steadily increasing. Seen in Fig 1, the growth in awarded master's and doctoral degrees were quite similar between 2005–2010: degrees increased by about 1,679 (30%) and 2,436 (26%) for doctoral and master's degrees, respectively. However doctoral degree growth stagnated after 2010, increasing by only 755 (9.4%) between 2010 and 2015. During those same years master's degrees increased by about 3,657 (18%). Bottlenecks in the research funding climate and pathway to the professoriate [4] potentially make a master's degree a more attractive path to pursue.

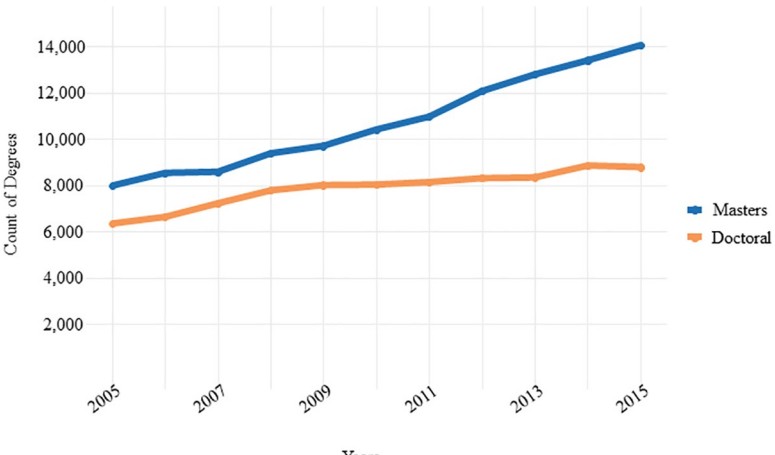

**Fig 1. Annual awarded biological science master's and doctoral degrees awarded in the United States.** Master's degree data was collected by the NSF from the Integrated Postsecondary Education Data System (IPEDS) Completions Survey, and doctoral degree data from the Survey of Earned Doctorates (SED) [10]. There were 6,367, 8,046, and 8,801 doctoral degrees awarded in the biological and biomedical sciences in 2005, 2010, and 2015; coincidingly, there were 7,988, 10,424, and 14,081 master's degrees awarded in the biological and biomedical sciences. SED collects data on the number and characteristics of individuals receiving research doctoral degrees from U.S. academic institutions. IPEDS surveys are administered to all Title IV-eligible universities and colleges in the United States and outlying territories, and is designed for collecting institution-level data from providers of postsecondary education. IPEDS considers a master's degree to require the successful completion of a program of study of at least the full-time equivalent of 1 academic year, but not more than 2 academic years, of work beyond the bachelor's degree [10, see technical notes for a full description of NSF data acquisition methods].

We conducted the first institutional analysis of standalone M.S. outcomes of 1,489 graduates from 15 of our Georgetown University Biomedical Graduate Education (BGE) M.S. programs. The 15 M.S. Programs analyzed are on average 30 credits each, require 1–1.5 years to complete full-time. With confounding definitions of "terminal" master's degrees prevalent among the literature, we refer to the 15 M.S. programs in this study as "standalone" throughout the text to emphasize that they are separate and distinct from our PhD programs. Our BGE master's program portfolio is housed in the Georgetown University Medical Center, along with our School of Medicine and School of Nursing and Health Studies. These M.S. degrees are conferred through Georgetown's Graduate School of Arts and Sciences, and consists of interdisciplinary science programs that involve collaborations across the basic and clinical sciences, policy, and industry. Our M.S. programs leverage cross disciplinary faculty expertise spanning our academic medical center, Lombardi Comprehensive Cancer Center, and Medstar Georgetown University Hospital, but they loosely cluster into data science, clinical and health sciences, basic science, and industry and policy programs. The 15 programs analyzed were: Biotechnology, Physiology and Biophysics, Biochemistry and Molecular Biology, Biostatistics, Biohazardous Threat Agents and Emerging Infectious Diseases, Integrative Medicine and Health Sciences, Integrative Neuroscience, Pharmacology, Tumor Biology, Bioinformatics, Microbiology and Immunology, Biomedical Science Policy and Advocacy, Clinical and Translational Research, Health Physics, and System's Medicine.

We find two main reasons why students are entering our master's programs: 1) to make them more competitive for advanced degree programs, such as research-based doctoral programs or medical and health professions programs, and 2) for workforce career advancement. Applicants to our master's programs are evaluated holistically by a program admissions committee, and are expected to have completed an undergraduate bachelors-level degree prior to matriculation, achieved a 3.0 minimum GPA, and submit a statement of purpose that details their educational background and career goals. Each graduate program's curriculum is uniquely designed to acquaint each cohort from the start, but program-specific curricular prerequisites are required to ensure that applicants have foundational knowledge pertinent to the rigorous coursework of the program.

While Professional Science Master's (PSM) program affiliation is administered by the Commission on Affiliation of PSM Programs [13], our standalone M.S. degree programs are not professional degrees. However, they share many similarities with PSM degree programs with regards to experiential learning opportunities and educational collaborations with industry employers. Our analysis of post-graduation outcomes for our standalone biomedical Master of Science (M.S.) degree programs suggests that their multidisciplinary research and academic curriculum provides students with knowledge and skills that translate directly to post-graduate career success. Directly after graduation, 69% of our M.S. graduates are employed in a variety of health-related environments; 28% pursue Ph.D. or medical/health profession programs. Graduates from these contemporary M.S. programs have unique and notable features in their workforce footprint and have changed the landscape of the biomedical workforce. Due to the variety and multidisciplinary nature of our programs, we see a robust footprint of career outcomes, primarily in research-driven job functions, across a myriad of industries. These findings present a new and contemporary paradigm for the role of the standalone Master of Science (M.S.) degree in preparing students for careers in the biomedical sciences.

## Methods

### Participants

We were able to track 1,204 (81%) of first destination outcomes out of our 1,489 M.S. graduates across our 15 BGE M.S. programs for all 5 years of the study. For our repeat measure

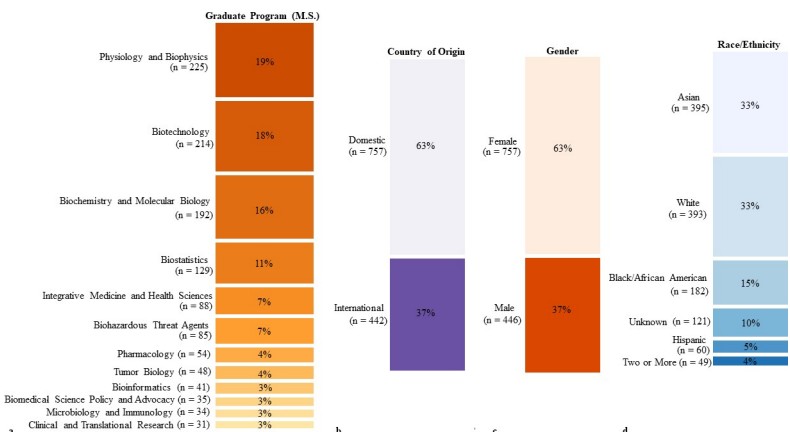

**Fig 2. Participant demographics.** Participant demographics were delineated according to the distribution of participants among graduate programs (a), country of origin (b), reported gender (c), and race/ethnicity (d). Only graduates where outcomes were found (n = 1204) are included. For gender, the non-binary (n = 1) category is not shown. M.S. Systems Medicine (n = 12), M.S. Health Physics (n = 9) and M.S. Integrative Neuroscience (n = 7) are not shown in (a). Five graduates did not have citizenship status recorded and are not included in (b). The American Indian or Alaska Native (n = 2) and Native Hawaiian/Other Pacific Islander Categories (n = 2) are not shown in (d).

study of 2016 and 2017 grads, there was a total of 412 (67%) out of 614 graduates where we found both first destination and two-year outcomes. An overall demographic breakdown of our population is shown in Fig 2, with the four largest M.S. graduate programs (by enrollment) studied representing 64% of the sample: Physiology and Biophysics, Biotechnology, Biochemistry and Molecular Biology, and Biostatistics (Fig 2A). We were able to find first destination outcomes for 757 (63%) domestic and 442 (37%) international graduates (Fig 2B). The majority of these international graduates, are from China, India, and Saudi Arabia. Sixty three percent of our 2014–2018 graduates identified as female and 37% identified as male (Fig 2C). Lastly, the racial and ethnic representation among graduates included in this study are 33% Asian, 33% White, 15% Black or African American, 5% Hispanic, and 14% who's race/ethnicity are unknown or who identified as two or more races. A total of four graduates (<1%) identified as American Indian/Alaskan Native or Native Hawaiian/Other Pacific Islander. Demographic categories were modeled after the Georgetown University Graduate School of Arts and Sciences admissions profile attributes.

## Analysis

We collected first destination (6 months after graduation) and repeat measure (2 years after graduation) placement outcomes for spring, summer, and fall 2014–2018 graduates. We took a two-stage approach to data collection: Direct outreach (survey and direct contact) and manual online search of remaining graduates through publicly available channels. Furthermore, for a subset of 2015–2018 graduates, we were able to link outcomes data to initial career goals gathered from a new student survey. All analyses in this paper were calculated with R-3.6.2 (The R Foundation; Vienna, Austria).

First destination and two-year outcomes were recorded as either employed, pursuing an advanced degree, volunteering, or seeking opportunities/unemployed. All employed graduates are categorized into an industry (job type) and a sector (job function). We audited the accuracy of our internal process for assigning sectors and industries by comparing our assignments to a subset of self-reported assignments from surveyed graduates. Our assignment accuracy was about 85% (the definitions we used for these assignments are provided for reference in

**Table 1. Definitions of sectors and industries used for categorization of employed graduates.**

| Industries | | Sectors | |
|---|---|---|---|
| **Academia** | Working for an academic institution in any capacity such as teacher, researcher, administrator, etc. | **Consulting** | Working as a consultant within any industry, providing analysis and solutions to the needs of a certain company or field. |
| **BioPharma** | Working in the areas of Biotechnology or Pharmaceuticals. | **Education** | Working to directly teach or educate as a tutor, lecturer, teaching assistant, etc. |
| **Business** | Working in the private sector for a company with no clear ties to any other industry listed here. | **Finance and Administration** | Working primarily on the logistics side of a company such as helping to run an office, facilitating operations, etc. |
| **Consulting** | Working for a consulting firm or in some consulting capacity. | **Marketing and Communications** | Working in a capacity that involves direct communication with other people such as patient coordinators, scientific liaisons, sales representatives, etc. |
| **Government and Policy** | Working for any federal office, policy think tank, political research company, etc. | **Medicine** | Working in a medical specific profession such as a scribe, dental assistant, etc. |
| **Healthcare** | Working in the fields of health sciences such as hospitals, health consulting, health advisors, etc. | **Regulatory Affairs** | Working to facilitate scientific work, such as a lab, meets all regulations and standards, federal or otherwise. |
| **Law** | Working in the field of law. | **Research** | Working in data gathering, analysis, and production in the clinical biosciences, policy, market trends, etc. |
| **Other** | For any industries that do not fit the categories above. | **Technology** | Working with computer programming, bioinformatics engineers/developers, etc. |
| | | **Other** | For any sectors that do not fit the categories above. |

Each first destination and two-year employment outcome was assigned to an industry and a sector.

Table 1). Therefore, we are confident in the fidelity of our reported sector and industry assignments for outcomes collected via manual search. Ethical approval has been granted for this study by the Georgetown University Institutional Review Board on November 27th, 2019. Reference number: STUDY00001411. Documentation of consent was not required because the data was analyzed retrospectively.

## Results

### M.S. graduates enter advanced degree programs

Our terminal M.S. degree programs provide advanced scientific and biomedical curricula to reinforce academic credentials and provide experiential learning opportunities. Graduates leverage this experience to enter the workforce or to commit to doctoral research degrees, health professions degrees, and other advanced degrees. Indeed, in our new matriculant student survey ($n$ = 305), 160 (52%) of BGE M.S. graduates from 2015–2018 indicated they were entering their M.S. program as a step towards pursuing an advanced degree such as a Ph.D. or M.D./D.O. degree. The remaining 145 (48%) graduates indicated career advancement as a primary motivator for pursuing their M.S. degree.

From 1,204 graduates across 15 BGE M.S. programs from 2014–2018, 1166 (97%) were either employed or in an advanced degree program (Fig 3). Of the 337 graduates in advanced degree programs, 116 (48%) enrolled in allopathic or osteopathic medicine (M.D. or D.O.) programs, and 115 (34%) enrolled in Ph.D. programs. Remaining graduates from the advanced degree pool pursued additional degrees in public health, business, law, pharmacy, nursing, dentistry, or allied health programs (Fig 2).

Outcomes were available for 269 of the graduates who completed our new student survey when they first started their M.S program, which allowed us to compare graduate intentions with outcomes (S1 Dataset). Interestingly, of the students who sought entry into advanced

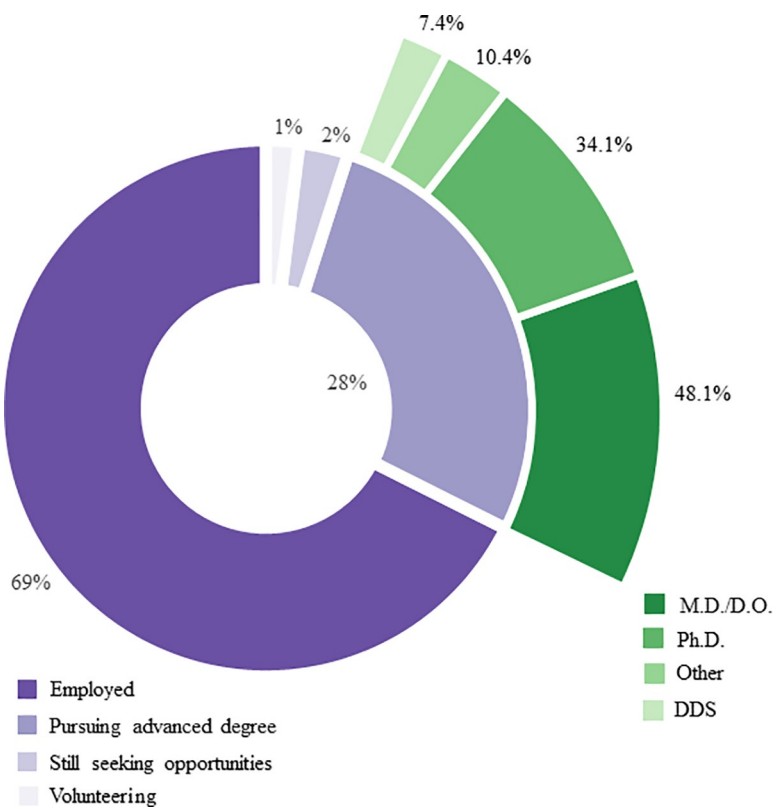

**Fig 3. First destination overall outcomes and advanced degrees.** This figure represents a snapshot of what 2014–2018 BGE graduates were doing after graduation ($N = 1489$, $n = 1204$). The percentages of degree types for graduates who were pursuing an additional advanced degree ($n = 337$) is also depicted. The category "Other" comprises M.S., M.P.H., D.M.D., M.B.A., M.S.N., D.P.M., J.D., P.A., P.A.-S., Pharm.D., D.C., D.V.M., M.P.P. and two unknown degree types.

degree programs as a primary motivator for enrolling in their M.S. program ($n = 139$), 56 (40%) of these students enrolled in an advanced degree program directly after graduating. Whereas, 75 (54%) graduates directly entered into the workforce. Of the students that sought career advancement as a primary motivator for enrolling in their M.S. program ($n = 130$), 113 (87%) were indeed employed, while 14 (11%) had decided to pursue an advanced degree.

From these findings it appears that overall graduates are leveraging skills from their program to enter the workforce and gain important experience needed for entry into advanced degree programs as they originally intended. The smaller proportion of graduates who sought entry into advanced degree programs as a primary motivator is perhaps in part explained by "gap years". Graduates may use this period for additional professional development through employment (i.e. gap year) to reapply for advanced degree programs if they were not admitted to these programs previously. We explore this possibility in a later section when we discuss the two-year outcomes of graduates who enter advanced degree programs after a gap year.

## M.S. graduates pursue broad scientific careers

Previous work has anticipated the use of professional biomedical master's degrees as a means for workforce placement and career advancement [8, 11–13]. Our data indicates that while a cohort of our M.S. degree graduates utilize their training to reinforce academic credentials for advanced degree programs, these terminal M.S. programs are also key stepping stones for

science-based positions in the broader workforce. The diverse and multidisciplinary curriculum of our M.S. programs uniquely allow graduates to apply their content expertise and leverage critical thinking and problem-solving skills to solve real world biomedical problems in health, science, business, and policy. Our program faculty forge partnerships with research groups and organizations to provide experiential, often research-based, opportunities to students. These short-term capstone projects allow students to gain practical workforce experience and transition to full-time positions after graduation.

We found that 829 (69%) of graduates from 2014–2018 directly entered the workforce after graduation (Fig 3). Seen in Fig 4A, the majority of these employed graduates took positions in academia, healthcare, biopharma, government, and policy roles. Notably, from Fig 3A it is clear the majority of graduates go into research-based work across all of these industries. More precisely, 488 (59%) of these employed graduates went into research. Also worth noting is that within academia, we see a footprint of alumni pursuing not only research and educator job functions, but also administrative and regulatory affairs roles.

Next, we analyzed the types of position titles these graduates pursue for an overall footprint of common roles of graduates within organizational employment structures. The scope of job titles depends on the overall talent management structure of an organization, and are thus difficult to compare from one industry/sector to another, or even between organizations in similar industries/sectors. Nonetheless, we are able to get a representative snapshot of the range of roles that graduates pursue. Seen in Fig 4B, most graduates place at assistant, associate, and analyst levels. Expertise-driven functional roles such as analyst, specialist, technician, scientist, and biostatistician suggest that alumni are making specialized contributions to their organizations, consistent with graduate training. We also observed senior and manager level roles, indicating that some graduates are also positioning themselves into managerial roles with a higher degree of responsibility.

## Employment and advanced degree outcomes of international M.S. degree recipients

We determined if there were differences in outcomes between domestic and international M.S. graduates. Overall, international and domestic graduates were either employed or pursuing advanced degrees postgraduation at similar proportions, with no significant difference found ($\chi^2 = 0.58$, $df = 1$, $P = 0.45$). Next, we determined the match between career intention at matriculation and actual post graduate outcome among domestic and international graduates.

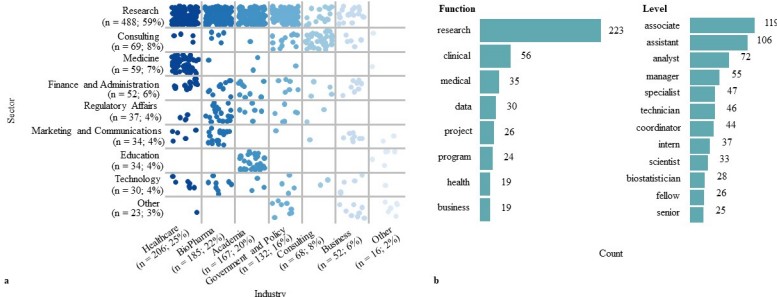

**Fig 4. First destination industry, sector, and position analysis.** a) a two-way table that plots the 829 employed 2014–2018 BGE graduates within each sector (job function) and industry (job type) for their first destination outcomes. Each graduate is represented as a dot, and the dots are colored by industry. The Law (n = 3) industry is excluded from this figure due to low counts. b) Counts of the twenty most frequent words within first destination position titles, separated into either a position function or position level category.

While the sample of domestic and international students (where both intention and outcomes data points were available) was too small to perform statistical analysis to compare the match trends between the two populations, we did observe that a majority of domestic and international students who sought career advancement as their motive for enrolling in their master's program were indeed employed after graduation (Fig 5A). However, of those who sought entry into an additional advanced degree program as their basis for enrolling in our master's programs, 37% of domestic students and 47% of international students were actually enrolled in an advanced degree program (such as M.D. or Ph.D. program) after completing our master's programs.

Out of the 314 international graduates employed post-graduation, 266 (84%) were able to find their employment within the United States (S1 Dataset). However, postgraduate industry proportions among the international alumni are significantly different from the proportions of employed domestic graduates ($\chi^2$ = 58.161, $df$ = 6, $P$ < 0.001). We observe that international graduates are less represented in government and policy roles ($n$ = 20, 6%) and more represented in academic ($n$ = 82, 26%) and biopharma ($n$ = 99, 31%) industries (Fig 5B).

Furthermore, the types of advanced degrees that domestic and international graduates pursue are quite different (Fig 5C). Out of the 106 international graduates pursuing advanced degrees, 84 (79%) were pursuing Ph.D.'s after completing their M.S. degree. While only 14 (13%) of international graduates were enrolled in medical degree programs (Fig 5C). These numbers are consistent with AAMC findings that 14% of international applicants enrolled in medical schools in the US in 2018 [14]. In contrast, we see a smaller proportion of advanced degree-seeking domestic graduates ($n$ = 229) pursuing Ph.D. degrees ($n$ = 31, 13%), while 146 (63.8%) were enrolled in medical programs (M.D. or D.O.). These differences in international and domestic graduates for Ph.D. and medical degree programs were statistically significant ($\chi^2$ = 117.8, $df$ = 1, $P$ < 0.001). Corroborating these placement trends, we also found that 89% ($n$ = 94) of international students who enrolled in advanced degree programs after completing their M.S. were enrolled in the following M.S. programs that contain research-intensive components or tracks: Biochemistry and Molecular Biology, Bioinformatics, Biostatistics,

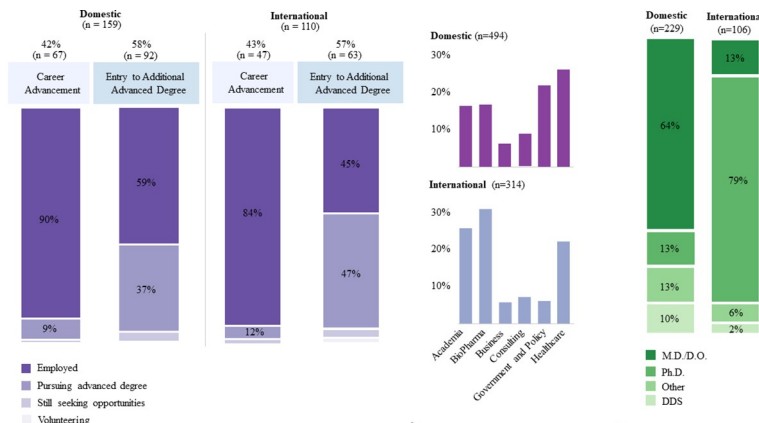

**Fig 5. Comparison of outcomes between domestic and international M.S. graduates.** a) A comparison of intent and outcome between domestic and international students. Outcomes of domestic students who sought career advancement (n = 67) or entry into an advanced degree program (n = 92) as a goal of enrolling in their M.S. program are shown on the left. Outcomes of international students who sought career advancement (n = 47) or entry into an advanced degree program (n = 63) as a goal of enrolling in their M.S. program are shown on the right. b) First destination industry percentages for domestic (country of origin is United States) (n = 494) and international (country of origin is not United States) (n = 314) and BGE graduates from 2014–2018. c) Distribution of advanced degree programs that domestic (n = 229) and international (n = 106) students enroll in after completing their M.S. degree.

Biotechnology, Microbiology and Immunology, Pharmacology, System's Medicine, and Tumor Biology (S1 Dataset). These programs provide requisite training for PhD programs including mentored research experiences, laboratory technical, and data analysis skills. Whereas 27% of domestic graduates (n = 61) who pursued advanced degree programs after completing their M.S. were graduates of these same programs.

## Analysis of gender and racial/ethnic distributions in employment and advanced degree outcomes

There were no notable gender-based differences in employment industry placements or advanced degree outcomes (S1 Fig), even though female-identifying graduates encompassed 63% of our sample population (Fig 2). Next, we analyzed racial and ethnic distributions among employment industries and advanced degree programs (Fig 6). Healthcare industry placements were proportionally overrepresented among Black and African American graduates. Furthermore, among Black/African American and Hispanic M.S. graduates who pursued advanced degrees, a majority (84% and 68%, respectfully) of graduates from those demographic groups enrolled in medical degree programs (M.D. or D.O.). Concordantly, we see that 59% (n = 143) of Black/African American and Hispanic graduates (n = 242 combined) among our sample population (graduates with trackable first destination outcomes) graduated from five healthcare/clinically-focused M.S. programs: Clinical and Translational Research, Integrative Medicine and Health Sciences, Pharmacology, Physiology and Biophysics, and Systems Medicine.

## Career observations from two years after graduation

We also conducted a repeat measure analysis of outcomes for 2016 and 2017 graduates. The aim of this additional repeat measure data collection was to observe possible career growth/ changes and delayed matriculation into advanced degree programs. There were a total of 412 (67%) graduates, out of the 614 graduates from 2016 and 2017, where *both* first destination and two year outcomes were found. Fig 7 depicts the overall career outcome changes for these graduates, with some apparent movements for originally employed graduates. For graduates that were employed six months after graduation (n = 282), two years after graduation 100

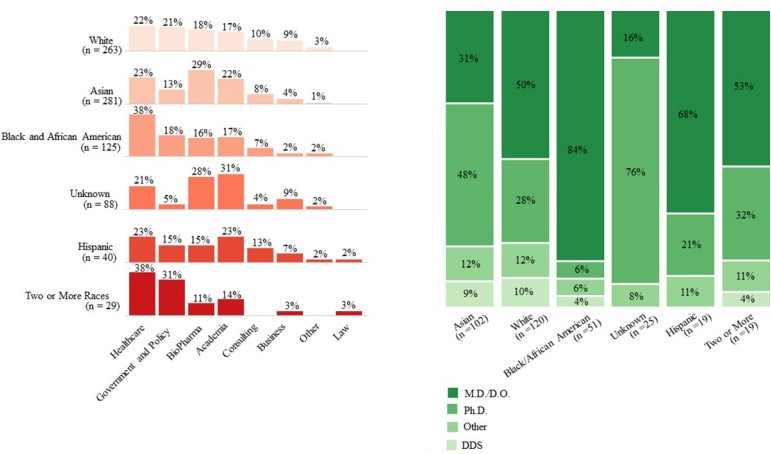

**Fig 6. First destination outcomes by race and ethnicity.** a) Industry placement distribution of employed graduates broken down by race and ethnicity. b) Race and ethnic representation among graduates who pursued advanced degree programs after completing their M.S. degree.

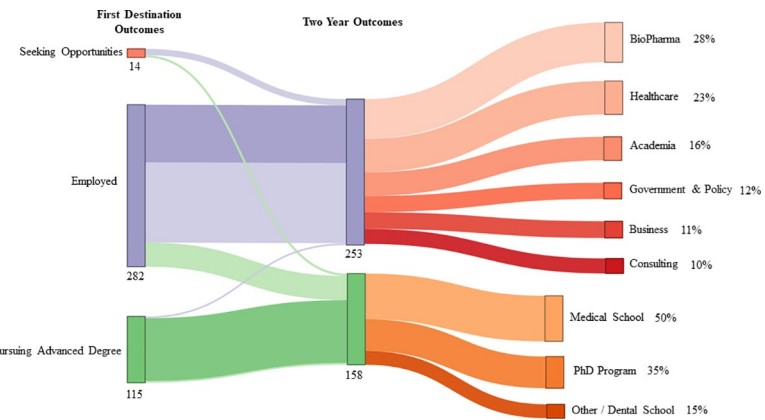

**Fig 7. Overall changes in outcomes.** Here, we present a comparison of first destination outcomes to two-year outcomes for BGE graduates of 2016 and 2017 ($N = 614$ total graduates; $n = 412$ outcomes found for first destination and two-year time points). The breakdown of industries and advanced degree programs on the right side represent the two-year outcomes. One additional graduate went from employed to unemployed.

(35%) of those graduates had no change in outcomes, 119 (42%) changed employers, 21 (7%) changed job functions at the same employer, and 42 (15%) graduates matriculated into advanced degree programs.

Among the graduates who changed positions ($n = 140$), either within the same organization or moved to a different organization, 55 (39%) changed sectors (job functions). Out of these 55 graduates who changed job functions, 30 (54%) moved out of the research sector. These movements out of the research sector suggest that a subset of graduates utilized primarily research functions as an entry point into their industry and then pursued more cross-functional responsibilities for their career advancement. Researchers are analytical and critical thinkers, and can learn and quickly adapt to new tasks and responsibilities.

We reiterate that position titles have varying breadth of responsibility across organizations, job sectors, and industries. However, we suspect that some markers of career growth would include movements out of job training positions (intern), as well as movements into positions of increased responsibility (managerial positions). Fig 8A compares first destination to two years position titles of individuals who remained in the workforce. Among those analyzed, we see movement out of intern positions and into longer term full-time jobs (Fig 8B). Internships appear to allow graduates to leverage their expertise from their advanced degree while

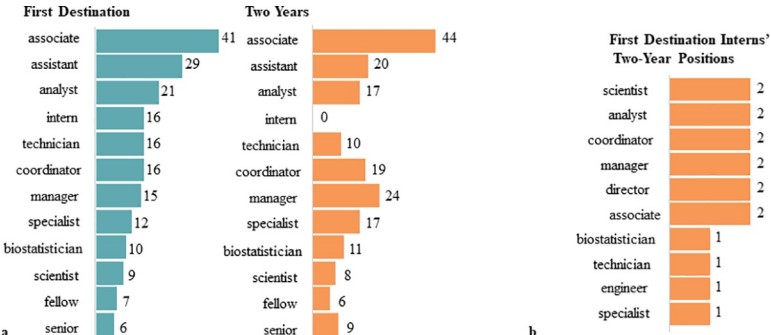

**Fig 8. Position title changes and advanced degrees.** a) Most frequent functional position title words b) Assessment of the professional movement of 16 interns two years post-graduation.

providing valuable work experience to increase their competitiveness for jobs that would require more independence. Internships also offer opportunities to preview the types of tasks and deliverables that would be expected of them. Moreover, in Fig 8A we can see an increase in manager-level positions suggesting that a subset of our M.S. graduates moved into positions of increased responsibility.

Students also seem to leverage the skills from their program to gain practical research, medical, and health science experience as a gateway toward an additional advanced degree program. Our M.S. programs are housed at our academic medical center and provide advanced biomedical curriculum and short-term research experiences across the basic and medical sciences. These experiences are of particular use for students interested in pursuing research and medical advanced degrees. For many doctoral and health professions advanced degree programs, applicants will need superior academic performance in addition to experiential activities in research and clinical settings. Depending on how much experience students may have before entering our M.S. program, or pursue during their M.S. program, we suspect that a cohort of graduates will need to pursue full-time employment to gain key experience and perspective pertinent to informing their decision to pursue an advanced degree and maximizing the competitiveness of their applications. Indeed, out of the 42 2016 and 2017 graduates who entered the workforce and enrolled in advanced degree programs two years later, 18 (43%) were pursuing Ph.D. degrees, 16 (38%) were pursuing M.D./D.O. degrees, with the remaining 8 (19%) pursuing other advanced degrees.

Out of this cohort of graduates that pursued advanced degree programs after being employed ($n = 42$), the two largest employment industries were Academia and Healthcare, which contained 15 (36%) and 11 (26%) graduates, respectively. We also found that graduates pursued placements in policy and business-related industries in the private sector. The complexity of the biomedical research and healthcare delivery and administration enterprises warrant cross-functional expertise in health, business, and policy; our M.S. graduates are pursuing employment opportunities that enrich their perspective in these areas before pursuing advanced degree studies.

We suspected that some students who were enrolled in advanced degree programs at the two-year time point would pursue professional experiences to strengthen their applications. We analyzed the job titles of graduates employed at the first destination time point and enrolled in advanced degree programs at two years, and found that 33% were placed in positions that gave them research and clinical exposure (n = 14), such as "Research Assistant/Associate" and "Medical Scribe". These positions are often experiential requisites for advanced degree programs such as Ph.D. and medical (M.D./D.O.) programs. Indeed, 10 graduates with an initial interest to pursue advanced degrees were employed at the first destination time point (33% of employed graduates with both a first destination and two-year outcome available), but were enrolled in advanced degree programs at the two-year time point (Fig 9A). Whereas, graduates with career advancement as an initial interest were still employed (96% of employed graduates with both a first destination and two-year outcome available), with more than half switching jobs and two employed (4%) alumni having entered advanced degree programs by the two-year timepoint (Fig 9B).

## Discussion

A M.S. degree in the biomedical sciences provides advanced coursework and experiential learning opportunities that imparts students with a cadre of skills necessary across the biomedical and broader scientific workforce. Furthermore, M.S. programs enhance students' credentials for highly competitive doctoral-level advanced degree programs and health professions

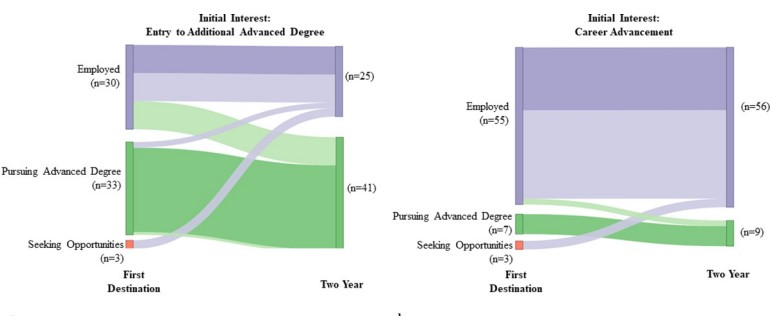

**Fig 9. Career changes by initial intentions.** This figure divides the 131 graduates who completed the new student survey and had both first destination and two-year outcomes found. a) Changes in first destination and two-year outcomes are shown for students initially intending to pursue an advanced degree after completing their M.S. b) Changes in first destination and two-year outcomes are shown for students initially seeking career advancement after completing their M.S.

programs. The curriculum of our programs plays a large role in the students that the program attracts and the subsequent post graduate outcome. Our programs are designed along a continuum of being primarily for workforce development or preparation for further advanced degree training. However, we see outcomes that reflect either extreme occurring in all of our M.S. programs, indicating the overall versatility of the biomedical M.S. degree.

## Industry exposure through M.S. degrees strengthens workforce development pipelines

Toward one end of the continuum, components of our industry focused master's programs such as Bioinformatics, Biostatistics, Biotechnology, Biomedical Science Policy and Advocacy, Biohazardous Threat Agents and Emerging Infectious Diseases, Clinical and Translational Research, and Health Physics leverage the expertise of industry leaders and practitioners in the curriculum. Experiential internships and capstone projects play a particularly important role in connecting students to professionals and industry mentors in their desired fields, which helps facilitate their career advancement. These projects also confer competencies that allow graduates to connect their science expertise with transferable business acumen, communications, and policy skills, which are important for closing a noted skills gap among graduates in the scientific labor market [15]. We also observed that 59% of graduates who were employed as a first destination outcome were performing a research function within their respective industry, indicating that our master's programs are filling an observed undersupply of graduate degree holders in the broad biomedical research workforce [16]. With many PhD programs having a focused attention on academic research, graduates from our 1–1.5-year industry-focused master's programs can readily combine scientific knowledge with an interprofessional skillset to competitively pursue a diverse set of career options.

International graduates in STEM fields can maintain their F1 student visa status for up to three years after graduation due to a 1-year Optional Practical Training (OPT) period and the associated two-year STEM extension. Yet most federal positions, as well as closely aligned positions in policy, require US citizenship or permanent residency. Short-term fellowship programs in the government and policy industry that do take international students often do not have opportunities for long-term sponsorship. Albeit competitive, the availability of sponsorship for advanced degree holders would offer our international graduates more security in the private and academic industries. There is a cap on the number of H1B visas available for the private sector to sponsor international employees, however nonprofit organizations such as

academic institutions are not currently subject to the H1B cap. Taken together, our findings indicate that while international students are underrepresented in government and policy roles in the US, they are able to find meaningful employment in other scientific and healthcare related nonprofit and private industries (Fig 5). However, extended tracking is needed to determine if our international graduates are able to secure long-term employment sponsorship.

While a majority of the students who enrolled in our more industry-focused programs tend to utilize this direct industry exposure to enter the workforce after graduating, a subset of these students entered medical degree programs with enriched perspectives on healthcare policy, drug development and regulatory policy, and global health security. The interdisciplinary training that students receive from industry-focused programs also provides valuable context for framing research questions and understanding the broader implications of their research findings for students who pursued PhD programs.

## M.S. degrees as a gateway toward advanced degree programs

On the other end of our M.S. program curriculum continuum, we have programs that prepare students to pursue advanced doctoral or health professions degrees. Our health science focused master's programs analyzed in this study, such as our Physiology and Biophysics, Integrative Medicine and Health Sciences, and System's Medicine programs prepare students for health professions advanced degree programs such as MD or DO programs. Our research focused programs such as Biochemistry and Molecular Biology, Pharmacology, Microbiology and Immunology, Integrative Neuroscience, and Tumor Biology provide students with advanced coursework in their discipline and robust research experiences to clarify their desire and expand their competitiveness to pursue PhD programs.

While 337 graduates across our programs pursue advanced degrees directly after completing our master's programs, we would expect nearly double that given the proportion of incoming students who stated an intention to pursue an advanced degree program as a reason for enrolling in their respective master's program. One reason for the discrepancy could be that some of these students are delayed in their matriculation into advanced degree programs. Indeed, our analysis shows that 42 tracked alumni who entered the workforce after completing their M.S. degree were enrolled into advanced degree programs by the two-year time point (Fig 7). There are several reasons why we suspect that these students are delayed in pursuing their advanced degrees. A desire to pursue professional development opportunities provides experiences that can address gaps in their applications, clarify their intentions to pursue advanced degrees, and secure wages to meet quality of life needs. A limitation of this study is that we are unsure if the delay to entering advanced degree programs is due to an unsuccessful application, particularly for students who are in the midst of an advanced degree application cycle while concurrently enrolled in our master's program. However, students who choose to apply to advanced degree programs after completing our master's degree would be entering admissions cycles that would lead to advanced degree program enrollments 1–2 years later for successful applicants, which aligns with our ability to capture those outcomes at our two-year data collection time point. In the future, a longitudinal study tracking the applicant journey for alumni pursuing advanced degrees would elucidate more of the nuanced dynamics of the application process.

The high proportion of international graduates attending Ph.D. programs indicates that international graduates are leveraging M.S. programs for curricular reinforcement and research experience to make themselves more competitive for Ph.D. programs. We speculate that while Ph.D. slots are still incredibly competitive, funding availability is amenable for

international student applicants since most biomedical Ph.D. programs cover tuition and stipends through institutional or external funding mechanisms. However international graduates are not eligible for federal financial aid and often have to otherwise prove that they will be financially responsible for all four years of typical medical school programs. Nonetheless, our international graduates do indeed enroll into medical school at a rate consistent with AAMC nationwide findings, which suggests a potential paradigm for international graduates to pursue a US-based biomedical M.S. degree to maximize their competitiveness for US-based medical schools.

## Perspectives and recommendations

By housing our terminal M.S. degrees in an academic medical center, we are able to provide an enriching learning experience for our graduate students by utilizing clinical and scientific expertise among faculty, research opportunities, and partnerships with cross-disciplinary academic departments, industry, and government agencies. Also, environments that provide specialized career and professional development opportunities and resources help students clarify their career goals, align their interests and skills, and broaden their networks [17]. Georgetown University Biomedical Graduate Education provides a full suite of career strategy resources including job search training, individual advising, document review for job applications and for advanced degree program essays, networking opportunities with alumni and employers, memberships to organizations, and leadership and management training. Furthermore, mentoring activities provide important career and psychosocial support [18]. We offer a structured near-peer mentoring program for master's students interested in advanced degree programs to learn from current Georgetown University M.D. or Ph.D. students. Institutions should continue to support and expand capacity for career development infrastructure and employer engagement to increase the portfolio of internship opportunities and help students leverage their skills to pursue their next professional endeavors.

Master's degree graduates have the advanced scientific expertise and cross-functional business, policy, and communications skills to help them navigate specialized jobs in the scientific workforce. Unemployment rates for master's degree recipients in the life sciences are lower than for bachelor's degree recipients [19]. Therefore, we expect to continue to see an increase in the development of these cross-disciplinary master's programs to prepare students for advanced degrees and career advancement. It is important to continue to track graduate outcomes in order to ensure that curriculum aligns with market demand. Within our own Georgetown University Biomedical Graduate Education portfolio, we've launched two new M.S. programs (Health Informatics and Data Science, and Medical Physics), our first M.A. program (Catholic Clinical Ethics) and our first professional master's program (Executive Master's in Clinical Quality Safety and Leadership) since completing this study.

Master's programs have transformed the biomedical workforce footprint by filling a human capital skills gap in the biomedical labor market, and should be financially supported through government and/or private sector sources. However, biomedical master's students are largely self-funded, with an average debt range of $20,000-$115,000 USD [20], a financial burden that is particularly difficult for students from economically disadvantaged communities. Our data indicate that our health science master's programs are a particularly important pipeline for underrepresented minority students to enter medical degree programs (Fig 6B). Therefore, expanded financial interventions to support master's studies are also necessary to broaden participation of students from marginalized communities into master's programs to continue to enrich and diversify a much-needed pipeline in health professions and academic research. The Georgetown University Biomedical Graduate Education Hoyas for Science Scholarship

currently provides a partial tuition scholarship to matriculants whose experience, when evaluated holistically, suggests that they are uniquely able to enrich the Georgetown University community. In the future, we envision an initiative that provides full funding support, specialized professional development, and mentorship activities to an annual cohort of master's students from marginalized and financially-disadvantaged backgrounds.

## Supporting information

**S1 Dataset. De-identified dataset of all alumni outcomes.**
(XLSX)

**S1 Fig. Postgraduate outcomes by gender.** a) Outcome status by gender. b) Industry placement distribution of employed graduates by gender. b) Gender representation among graduates who pursued advanced degree programs after completing their M.S. degree.
(TIF)

## Acknowledgments

The authors wish to thank Dr. Barbara S. Bregman and Dr. Kyle Divito for their critical reading of this article.

## Author Contributions

**Conceptualization:** Caleb C. McKinney.

**Data curation:** Jack Hart, Caleb C. McKinney.

**Formal analysis:** Jack Hart, Caleb C. McKinney.

**Investigation:** Caleb C. McKinney.

**Methodology:** Jack Hart, Caleb C. McKinney.

**Project administration:** Jack Hart, Caleb C. McKinney.

**Resources:** Caleb C. McKinney.

**Supervision:** Caleb C. McKinney.

**Visualization:** Jack Hart, Caleb C. McKinney.

**Writing – original draft:** Jack Hart, Caleb C. McKinney.

**Writing – review & editing:** Jack Hart, Caleb C. McKinney.

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
