## [Decision Letter · Decision Letter 0]

1 Oct 2020

PONE-D-20-19103

An Institutional Analysis of Graduate Outcomes Reveals a Contemporary Workforce Footprint for Biomedical Master’s Degrees

PLOS ONE

Dear Dr. McKinney,

Thank you for submitting your manuscript to PLOS ONE. After careful consideration, we feel that it has merit but does not fully meet PLOS ONE’s publication criteria as it currently stands. Therefore, we invite you to submit a revised version of the manuscript that addresses the points raised during the review process.

We look forward to receiving your revised manuscript.

Kind regards,

Amy Prunuske

Academic Editor

PLOS ONE

Additional Editor Comments:

Thank you for sharing your work. I encourage you to address the reviewers' comments. In particular, there were several suggestions that would strength the discussion. It would also be helpful to describe the demographics of your population, selection criteria for the programs, and average debt. Similar to the analysis that was done with the international and national graduates it would be interesting to know if there were any differences in the outcomes by gender or race.

Journal Requirements:

Reviewers' comments:

Reviewer's Responses to Questions

**Comments to the Author**

1. Is the manuscript technically sound, and do the data support the conclusions?

Reviewer #1: Yes

Reviewer #2: Partly

2. Has the statistical analysis been performed appropriately and rigorously? 

Reviewer #1: Yes

Reviewer #2: Yes

3. Have the authors made all data underlying the findings in their manuscript fully available?

Reviewer #1: No

Reviewer #2: No

4. Is the manuscript presented in an intelligible fashion and written in standard English?

Reviewer #1: Yes

Reviewer #2: Yes

5. Review Comments to the Author

Reviewer #1: This paper adds to the current work on biomedical science career paths and navigation and importantly focuses on programs for MS graduates, rather than PhDs. This is a segment that is missing in the current literature, and the authors do a nice job of describing the project and outlaying the results.

Some areas to revise for clarity--

1. Abstract line 3: "one to two year" instead of "one-two"

2. In some areas, it shows "masters degrees" or "Master's degrees" or "master's degrees"

3. Similarly, there are a lot of words that are often hyphenated "full-time" "short-term" "cross-functional" "long-term"

4. Introduction paragraph 1 sentence 2, "most" and "others" could be more specific.

5. Introduction paragraph 3 lines 1-3, need citation. If it is related to the rest of the paragraph, you could move up the [9] to end of the first sentence and then have a transitional word for sentence 2 to let the reader know these are tied together.

6. Tried to access citation 10 for question re: page 4, but the link in reference list goes to error. Question is about the NSF data--is this master's degrees in total or on the way to doctoral degree. Many institutions are now awarding MS on the way to PhD or if the student exits the program where previously they were only offering a PhD for full completion.

7. The link for citation 11 also does not open.

8. I would argue this is not a longitudinal study as there are two data collection points over a short time. Two years seems like a long time when conducting a study, but when it involves career paths, it is a very short time. This seems more like a repeated measures study. One should not expect a graduate to have changed careers (sectors or industries drastically or to have increased significantly in job responsibility) within 2 years of obtaining a master's degree. Where graduates initially go is key, as are useful skills obtained from graduate program. The point on p 10 re: using terminal MS programs as stepping stones is central to the authors' points. I would add that the workforce experience or capstone projects also allow the students to connect and network with industry leaders, which paves the way for the transition to a position and career path or perhaps connecting with someone who may fill the role of a mentor to the new graduate.

9. p6, line 6, more clear to say "(the definitions we used for these assignments are provided for reference in Table 1)."

10. In results section, chi square symbol is replaced with box.

11. A few places throughout need comma or have extra space.

12. p11, last line of first paragraph, replace "education" with "educator" or "teaching" for clarity

13. p12, last paragraph, is it that the graduates used jobs with research functions intentionally as gateway to other responsibilities or are those the only jobs a graduate with MS can access and then needs to gain experience before finding another job?

14. p13, line 4 "managerial"

15. p13, end of paragraph 1, internships give graduate opportunity to explore career but do not ensure they find career aligning with passions and goals. This is in part because internships are brief and have less responsibility and lower range of job tasks or that the student is still determining preferences and path. Also, jobs immediately after graduation or ones offered as internships are not often the ones students envision as long-term career.

16. p13, paragraph 2, line 6, "-level" not needed

17. p14, "were" not needed at end of next to last line of 1st paragraph. "with the remaining 8 pursuing...."

18. p14, paragraph 2, line 3, "of" not needed before "graduates"

19. p15, paragraph 2 line 1, add "visa" to "F1 student visa status"

20. Discussion is very brief. Were there any limitations to this study? More information on what to study next or important considerations needed. A point was made about this being important due to self-funding, but I would argue that if self-funded, it is important to the student this work be done but should be important to federal funding agencies and institutions if those are funding it. Additionally some work and programs are arguing the case that master's prepared graduates perform essential STEM functions and education should be financially supported. Last line about institutions needing to provide career development infrastructure and employer engagement--there is some research about the many institutions that do provide career development opportunities and resources. It may need to be revised or increased, but this statement indicates it is not being done.

21. p17 top line, '"is in line with" to "aligns with"

Reviewer #2: This manuscript explores the significant issue around outcomes for students graduating with an MS degree from an academic medical center. In an era when many PhD graduates are pursuing a range of career outcomes, it begs the question whether for some of these careers, advanced study at the MS level would be a more direct route to the goal. This analysis helps to address this question, but a few improvements are suggested to the authors.

• The manuscript begins with a statement of the results in the initial sentence. It would be better to frame the context of this work, the types of degrees earned in the biological sciences, and to define the terms terminal (vs. professional) MS degree.

o Use of the word terminal implies PhD students who have left their doctoral programs and are now pursuing an MS. Is that the intended meaning? That would also influence the analyses looking at long term career goals.

o Along these lines, some background of the 15 included degree-granting programs and the typical outcomes for which they try to prepare their graduates would be useful. Surely, not all MS degrees are created equal, with some focusing more on career outcomes immediately after graduation and others positioning graduates for further advanced study.

• Figure 1 is drawn from NSF data, and readers are pointed toward NSF technical notes, but the reviewer advises inclusion of a brief statement indicating what types of institutions were included and what subfields are represented.

• On page 7, the authors describe outcomes for 337 graduates across the programs with respect to the degrees they are currently pursuing. Given the stated goals upon matriculation, we would expect closer to 600 graduates in advanced degree programs. Reasons for the large discrepancy in the initial goals vs. the actual outcomes should be explored further in the discussion.

• Some aspects of interpretation appear in the results and would be a better fit for the discussion, e.g. page 8 (last paragraph on page) and page 10 (top of page).

• Page 9: change “very proportionate” to “proportional.”

• Statistical outcomes – letter conversion introduced an error. Should the square boxes be replaced by chi-squared?

• Do domestic and international graduates represent the same programs? How would this track back to the reviewer request for more information on goals for the graduates of each program? In other words, is the higher proportion of international grads heading to PhD programs in line with initially stated goals upon matriculation or program selection for MS programs that are more research-focused?

• It would be helpful to know what sorts of career mentoring is available to students in each of the programs. It is briefly mentioned in a very specific case at the bottom of page 10, but further information on the sorts of assistance provided to the students and in which career domains would be helpful. This may help readers understand if certain outcomes are supported to a greater extent than others.

• For the analysis looking at two year career status, can this be tracked back to initially stated career goals at the time of matriculation? Or were those data not yet collected for that cohort? Can it be provided for some of the graduates?

o Also for these graduates who are in an advanced degree program after two years but not initially, is it possible to determine if there was an unsuccessful application cycle in between, thereby necessitating the gap year?

• Figure 5c is probably adequate as text, but if there are no limits on figures, do not object to keeping it.

• Recommend in the discussion spending some time thinking about the transferability of the findings and the relevance of this work in the overall context of the biomedical workforce.

• The data are mentioned as being available with restrictions, and are provided as a supplemental dataset, but I defer to the editorial staff to determine if the restricted dataset meets publication requirements.

Overall, these are interesting findings, and with a bit of clarification of terms and context as well as some of the more nuanced analyses suggested above, this work is likely to be of interest to the readers of PLoS One.

6. PLOS authors have the option to publish the peer review history of their article (what does this mean?). If published, this will include your full peer review and any attached files.

Reviewer #1: **Yes: **Andrea M Zimmerman

Reviewer #2: No

---

## [Author Response · Author response to Decision Letter 0]

1 Nov 2020

It is with great enthusiasm that we address the Editor and Reviewer comments for our manuscript: An Institutional Analysis of Graduate Outcomes Reveals a Contemporary Workforce Footprint for Biomedical Master’s Degrees. We believe that the revisions have greatly improved the paper, and we are so appreciative to you and to the reviewers for your thoughtful feedback. I would like to take this opportunity to address your and the reviewer’s comments below. My response to each comment is in bold. 

We have carefully reformatted the manuscript to meet the PLOS ONE style requirements. In order to meet the style requirements and the reviewer comments, several figures were either created or modified to align with their reference in the text. We’ve also made additions to and rearrangements of the text to accommodate the Editor and Reviewer requests, and to adhere to the PLOS ONE style requirements. 

Additional Editor Comments:

1. In particular, there were several suggestions that would strength the discussion. 

We were happy to address those recommendations in our discussion section. We detail our responses to each of those points below. 

2. It would also be helpful to describe the demographics of your population

We’ve added a participants subsection in the Materials and Methods section where we describe the demographics of our population. Furthermore, we’ve generated a new figure 2 that illustrates the demographic breakdowns. 

3. Selection criteria for the programs, 

We have added selection criteria for admissions to our graduate programs in the Introduction section of the revised manuscript. 

4. Average debt. 

We analyzed average debt incurred by graduates of biomedical masters programs at private nonprofit institutions in the United States using publicly available datasets from the College Scorecard. We discuss a range of mean debt incurred in the discussion section and added a link to the publicly available raw data set in the references section. We’ve also tied in this discussion point with addressing item #20 from Reviewer 1 below. 

5. Similar to the analysis that was done with the international and national graduates it would be interesting to know if there were any differences in the outcomes by gender or race.

We have added a new figure 6 and accompanying addition to the results and discussions sections of the text that describe demographic differences among the outcomes. 

Reviewer #1 Comments:

1. Abstract line 3: "one to two year" instead of "one-two"

We’ve made this correction.

2. In some areas, it shows "masters degrees" or "Master's degrees" or "master's degrees"

We have standardized the use of master’s degree(s) throughout. When beginning a sentence or referring to a specific graduate program, we use Master’s. 

3. Similarly, there are a lot of words that are often hyphenated "full-time" "short-term" "cross-functional" "long-term"

We went through the manuscript and hyphenated those words throughout. 

4. Introduction paragraph 1 sentence 2, "most" and "others" could be more specific.

We’ve replaced “most” and “others” with actual metrics from our data. Also, this paragraph is now the last paragraph of the introduction section to accommodate reviewer feedback to add more background context to our findings in the introduction section. 

5. Introduction paragraph 3 lines 1-3, need citation. If it is related to the rest of the paragraph, you could move up the [9] to end of the first sentence and then have a transitional word for sentence 2 to let the reader know these are tied together.

We’ve incorporated this recommendation into that paragraph to more clearly tie the remarks in that paragraph to the indicated reference. 

6. Tried to access citation 10 for question re: page 4, but the link in reference list goes to error. Question is about the NSF data--is this master's degrees in total or on the way to doctoral degree. Many institutions are now awarding MS on the way to PhD or if the student exits the program where previously they were only offering a PhD for full completion.

We have corrected the link. For the IPEDS data set, awarded master's degrees are based on the completion of a program. In the Figure 1 legend, we have clarified how NSF defines awarded master’s degrees. 

7. The link for citation 11 also does not open.

We have corrected the link url.

8. I would argue this is not a longitudinal study as there are two data collection points over a short time. Two years seems like a long time when conducting a study, but when it involves career paths, it is a very short time. This seems more like a repeated measures study. One should not expect a graduate to have changed careers (sectors or industries drastically or to have increased significantly in job responsibility) within 2 years of obtaining a master's degree. Where graduates initially go is key, as are useful skills obtained from graduate program. The point on p 10 re: using terminal MS programs as stepping stones is central to the authors' points. I would add that the workforce experience or capstone projects also allow the students to connect and network with industry leaders, which paves the way for the transition to a position and career path or perhaps connecting with someone who may fill the role of a mentor to the new graduate.

We agree with the reviewer that this is indeed a repeat measure study, and we have made the necessary modifications to the text. We also discuss longitudinal studies that we plan to pursue for future followup work in the Discussion section. Furthermore, the points made by the reviewer regarding internships as networking opportunities and mentored experiences are well received and have been incorporated into the discussion. 

9. p6, line 6, more clear to say "(the definitions we used for these assignments are provided for reference in Table 1)."

Thank you for this recommendation, we have amended the text accordingly. 

10. In results section, chi square symbol is replaced with box.

We will ensure that our symbols carry over in the final submission. 

11. A few places throughout need comma or have extra space.

We have performed a find function for these doubles spaces and have replaced them with a single space. 

12. p11, last line of first paragraph, replace "education" with "educator" or "teaching" for clarity

We have replaced “education” with “educator” for clarity. 

13. p12, last paragraph, is it that the graduates used jobs with research functions intentionally as gateway to other responsibilities or are those the only jobs a graduate with MS can access and then needs to gain experience before finding another job?

Great question. While our study did not assess first destination job satisfaction of employed graduates, we have explored the workforce implications of a cross-industry research footprint more in the discussion. We interpreted the research function footprint across industries as a valuable asset to the labor market. 

14. p13, line 4 "managerial"

We have replaced “manager level” with “managerial”)

15. p13, end of paragraph 1, internships give graduate opportunity to explore career but do not ensure they find career aligning with passions and goals. This is in part because internships are brief and have less responsibility and lower range of job tasks or that the student is still determining preferences and path. Also, jobs immediately after graduation or ones offered as internships are not often the ones students envision as long-term career.

Thank you for these important insights. While we do think that internships are chosen in part due to career passions, we acknowledge that they also can be a pivotal step in the career exploration and goal formation process. We have modified the language of that paragraph of the results section to reflect the role of internships in career exploration. We have also explored the role of internships in more depth in the Discussion section. 

16. p13, paragraph 2, line 6, "-level" not needed

Thank you, “-level” has been removed. 

17. p14, "were" not needed at end of next to last line of 1st paragraph. "with the remaining 8 pursuing...."

We’ve made this correction. 

18. p14, paragraph 2, line 3, "of" not needed before "graduates"

We’ve made this correction. 

19. p15, paragraph 2 line 1, add "visa" to "F1 student visa status"

We’ve made this correction. 

20. Discussion is very brief. Were there any limitations to this study? More information on what to study next or important considerations needed. A point was made about this being important due to self-funding, but I would argue that if self-funded, it is important to the student this work be done but should be important to federal funding agencies and institutions if those are funding it. Additionally some work and programs are arguing the case that master's prepared graduates perform essential STEM functions and education should be financially supported. Last line about institutions needing to provide career development infrastructure and employer engagement--there is some research about the many institutions that do provide career development opportunities and resources. It may need to be revised or increased, but this statement indicates it is not being done.

Per the reviewers recommendations, we have added study limitations to the discussion. We have also explored the implications of our findings for funding opportunities. Lastly, regarding career development opportunities, we applaud the efforts that institutions have made. We were hoping that our findings would reinforce the need to sustain and increase capacity for these important institutional resources. We have modified the discussion to discuss our own career development programs and resources, acknowledge the efforts of other institutions, and to advocate for the value of institutional investment in these important career preparation resources. 

21. p17 top line, '"is in line with" to "aligns with"

We’ve made this correction. 

Reviewer #2 Comments:

1. The manuscript begins with a statement of the results in the initial sentence. It would be better to frame the context of this work, the types of degrees earned in the biological sciences, and to define the terms terminal (vs. professional) MS degree.

We moved the first paragraph to the last paragraph in the introduction section in order for it to have better context. We also provide clarity that our master’s degrees analyzed in this manuscript are standalone master’s degree programs that are distinct from our PhD programs. We also clarify that our standalone masters programs share many similarities with Professional Science Masters degrees with regards to experiential learning opportunities and educational collaborations with industry employers. 

2. Use of the word terminal implies PhD students who have left their doctoral programs and are now pursuing an MS. Is that the intended meaning? That would also influence the analyses looking at long term career goals.

Thank you for bringing up this point. Our master’s programs are distinct from our PhD programs, and we have edited the introduction to clarify that distinction.

3. Along these lines, some background of the 15 included degree-granting programs and the typical outcomes for which they try to prepare their graduates would be useful. Surely, not all MS degrees are created equal, with some focusing more on career outcomes immediately after graduation and others positioning graduates for further advanced study.

We have modified the discussion to delineate some of these program-specific differences in outcomes based on the goals of the programs, to better contextualize the findings. 

4. Figure 1 is drawn from NSF data, and readers are pointed toward NSF technical notes, but the reviewer advises inclusion of a brief statement indicating what types of institutions were included and what subfields are represented.

Per reviewer recommendation, we have updated the figure 1 figure legend to include a description of institutions and the subfields represented. 

5. On page 7, the authors describe outcomes for 337 graduates across the programs with respect to the degrees they are currently pursuing. Given the stated goals upon matriculation, we would expect closer to 600 graduates in advanced degree programs. Reasons for the large discrepancy in the initial goals vs. the actual outcomes should be explored further in the discussion.

We take the reviewers suggestion and explore the discrepancy between initial goals vs actual outcomes in the Discussion section. We are able to show that the discrepancy is in part resolved by delayed matriculations into advanced degree programs after entering the workforce first after finishing their masters degree. 

6. Some aspects of interpretation appear in the results and would be a better fit for the discussion, e.g. page 8 (last paragraph on page) and page 10 (top of page).

Per reviewer recommendation, we have moved both of those indicated sections to the discussion. 

7. Page 9: change “very proportionate” to “proportional.”

We’ve made this correction. 

8. Statistical outcomes – letter conversion introduced an error. Should the square boxes be replaced by chi-squared?

Thank you, we will ensure that our symbols carry over in the final submission. 

9. Do domestic and international graduates represent the same programs? How would this track back to the reviewer request for more information on goals for the graduates of each program? In other words, is the higher proportion of international grads heading to PhD programs in line with initially stated goals upon matriculation or program selection for MS programs that are more research-focused?

While we cannot discern from our incoming student survey the type of additional advanced degree program that incoming M.S. students were interested in pursuing post graduation, we can confirm that there is indeed a higher proportion of international students who graduated from our research-focused M.S. programs to pursue an additional advanced degree (89% international vs. 27% domestic). We’ve added this analysis to the results section. 

10. It would be helpful to know what sorts of career mentoring is available to students in each of the programs. It is briefly mentioned in a very specific case at the bottom of page 10, but further information on the sorts of assistance provided to the students and in which career domains would be helpful. This may help readers understand if certain outcomes are supported to a greater extent than others.

We have modified the discussion to discuss our own career development programs and resources, acknowledge the efforts of other institutions, and to reinforce the value of institutional investment in these important career preparation resources. 

11. For the analysis looking at two year career status, can this be tracked back to initially stated career goals at the time of matriculation? Or were those data not yet collected for that cohort? Can it be provided for some of the graduates?

We have addressed this request with a new figure 9 and accompanying modifications to the results and discussion sections to accommodate the new analysis. 

12. Also for these graduates who are in an advanced degree program after two years but not initially, is it possible to determine if there was an unsuccessful application cycle in between, thereby necessitating the gap year?

We are unable to determine this explicitly from our data, but we have added more interpretation of our findings in the discussion section to address this question. A limitation of our survey design is that we do not inquire about where students are in the application process for advanced degree programs. In a future study, we will map the applicant journey more precisely.

13. Figure 5c is probably adequate as text, but if there are no limits on figures, do not object to keeping it.

Thank you for this recommendation. We agree with the reviewer and have removed panel “c” from what is now Figure 8.

14. Recommend in the discussion spending some time thinking about the transferability of the findings and the relevance of this work in the overall context of the biomedical workforce.

We have made extensive revisions to the Discussion section to reflect more on the implications of our findings. 

15. The data are mentioned as being available with restrictions, and are provided as a supplemental dataset, but I defer to the editorial staff to determine if the restricted dataset meets publication requirements.

Restriction is necessary for this disaggregated dataset in order to maintain student privacy standards. Of note, to accommodate the additional analyses performed for this revision, we have added a column to the S1 Dataset to distinguish domestic and international students.

---

## [Decision Letter · Decision Letter 1]

17 Nov 2020

An Institutional Analysis of Graduate Outcomes Reveals a Contemporary Workforce Footprint for Biomedical Master’s Degrees

PONE-D-20-19103R1

Dear Dr. McKinney,

We’re pleased to inform you that your manuscript has been judged scientifically suitable for publication and will be formally accepted for publication once it meets all outstanding technical requirements.

Kind regards,

Amy Prunuske

Academic Editor

PLOS ONE

Additional Editor Comments (optional):

Thank you for your revised submission and your thoughtful responses to the concern. I am happy to accept your manuscript. The Reviewer did note that you have not provide all of the data underlying your manuscript. Please determine if it would be appropriate to share this data as supporting information.

Reviewers' comments:

Reviewer's Responses to Questions

**Comments to the Author**

1. If the authors have adequately addressed your comments raised in a previous round of review and you feel that this manuscript is now acceptable for publication, you may indicate that here to bypass the “Comments to the Author” section, enter your conflict of interest statement in the “Confidential to Editor” section, and submit your "Accept" recommendation.

Reviewer #1: All comments have been addressed

2. Is the manuscript technically sound, and do the data support the conclusions?

Reviewer #1: Yes

3. Has the statistical analysis been performed appropriately and rigorously? 

Reviewer #1: Yes

4. Have the authors made all data underlying the findings in their manuscript fully available?

Reviewer #1: No

5. Is the manuscript presented in an intelligible fashion and written in standard English?

Reviewer #1: Yes

6. Review Comments to the Author

Reviewer #1: (No Response)

7. PLOS authors have the option to publish the peer review history of their article (what does this mean?). If published, this will include your full peer review and any attached files.

Reviewer #1: **Yes: **Andrea M Zimmerman

---

## [Editor Report · Acceptance letter]

26 Nov 2020

PONE-D-20-19103R1 

An Institutional Analysis of Graduate Outcomes Reveals a Contemporary Workforce Footprint for Biomedical Master’s Degrees 

Dear Dr. McKinney:

I'm pleased to inform you that your manuscript has been deemed suitable for publication in PLOS ONE. Congratulations! Your manuscript is now with our production department. 

Kind regards, 

on behalf of

Dr. Amy Prunuske 

Academic Editor

PLOS ONE